# Anti-phage islands force their target phage to directly mediate island excision and spread

Amelia C. McKitterick[1] & Kimberley D. Seed [1,2]

*Vibrio cholerae*, the causative agent of the diarrheal disease cholera, is antagonized by the lytic phage ICP1 in the aquatic environment and in human hosts. Mobile genetic elements called PLEs (phage-inducible chromosomal island-like elements) protect *V. cholerae* from ICP1 infection and initiate their anti-phage response by excising from the chromosome. Here, we show that PLE 1 encodes a large serine recombinase, Int, that exploits an ICP1-specific protein as a recombination directionality factor (RDF) to excise PLE 1 in response to phage infection. We show that this phage-encoded protein is sufficient to direct Int-mediated recombination in vitro and that it is highly conserved in all sequenced ICP1 genomes. Our results uncover an aspect of the molecular specificity underlying the conflict between a single predatory phage and *V. cholerae* PLE and contribute to our understanding of long-term evolution between phage and their bacterial hosts.

---

[1] Department of Plant and Microbial Biology, University of California, Berkeley, 111 Koshland Hall, Berkeley, CA 94720, USA. [2] Chan Zuckerberg Biohub, San Francisco, CA 94158, USA. Correspondence and requests for materials should be addressed to K.D.S. (email: kseed@berkeley.edu)

 1

Bacteria and phage are in a constant battle for survival that shapes the evolution of both populations over time[1–3]. Bacteria evolve to overcome the challenges of phage predation through a variety of mechanisms, including phage receptor variability, DNA degradation mechanisms (e.g. restriction–modification systems and CRISPR–Cas systems), and phage inducible chromosomal islands (PICIs)[4,5]. While the former programs are deployed before phage infection to prevent the phage from completing its lifecycle, PICIs respond to a specific cue expressed by a target phage upon infection. PICIs are typically thought of as phage parasites that provide a fitness advantage to their host bacterium by limiting phage proliferation[6] and by carrying important virulence genes, such as those found in *Staphylococcus aureus*[7]. Upon induction, PICIs excise from the host chromosome, replicate, and redirect the target phage's packaging machinery, ultimately inhibiting phage replication and enabling PICI transduction[8–12].

*Vibrio cholerae*, the causative agent of the diarrheal disease cholera, interacts with predatory phages in the aquatic environment as well as in the human host[13,14], leading to speculation that phages influence cholera epidemic dynamics[15,16]. In particular, the lytic phage ICP1 has been recovered from cholera patient stool and water samples over at least 12 years in Bangladesh[17–19] and is consequently considered a persistent predator of epidemic *V. cholerae* in this region. In response to persistent ICP1 predation, *V. cholerae* has acquired the phage-inducible chromosomal island-like element (PLE)[18]. The five PLEs identified in *V. cholerae* isolates spanning a >60-year collection period exhibit a common, ICP1-dependent response, which initiates with the integrated PLE excising from the chromosome and circularizing[20] (Fig. 1a). PLE excision facilitates mobilization as PLEs are transduced following ICP1 infection, permitting their spread to *V. cholerae* recipients[20]. Although the underlying mechanisms are not known, PLEs block the phage replication program in what appears to be an ICP1-specific manner, as other tested cholera phages do not stimulate PLE circularization and are not blocked by PLEs[20]. Further, protection against ICP1 is absolute—PLE+ cells produce no progeny phage. The transmission costs imposed by PLEs are a significant burden to ICP1 in nature, as some ICP1 isolates have acquired a CRISPR–Cas system to target PLEs, which allows ICP1 to persist in spite of PLE[18].

Here, due to the apparent specificity and conservation of PLE circularization in response to ICP1, we set out to characterize the mechanism governing this response. We attribute PLE mobility to a PLE-encoded large serine recombinase, Int, that hijacks a specific phage protein, which we refer to as PexA, to direct PLE excision during ICP1 infection. We validate the functionality and specificity of this unique recombination system, in which the recombinase and recombination directionality factor (RDF) are encoded in separate genomes. Additionally, we show that PexA is also hijacked to trigger the excision of PLEs found in *V. cholerae* isolates recovered decades ago, highlighting the continued arms race that shapes long-term evolutionary trajectories of ICP1 and epidemic *V. cholerae*.

## Results

### PLE−encoded Int is necessary for PLE circularization.
PLE encodes a gene product (Int) with an N-terminal serine recombinase domain and a large C-terminus containing a putative zinc ribbon domain and coiled–coiled motif characteristic of large serine recombinases (LSRs)[21]. Typically found in temperate phages, LSRs have the ability to catalyze recombination between attachment (*att*) sites[22–24]. Only the LSR is required to catalyze recombination between episomal (*attP*) and chromosomal (*attC*) sites, leading to integration of the episome into the host

chromosome. To reverse this process, an RDF is required to physically interact with the LSR and direct the LSR to recombine the left (*attL*) and right (*attR*) attachment sites, resulting in excision of DNA between these sites (Fig. 1b). In addition to what has been documented regarding PLE circularization following ICP1 infection under laboratory conditions[18,20], we note that ICP1-dependent PLE circularization can be detected in cholera patient stool (Fig. 1c), underscoring that *V. cholerae* PLE responds to ICP1 infection during disease in humans.

To determine if Int plays a role in PLE circularization during ICP1 infection, PLE 1 Δ*int* was challenged with ICP1. Unlike wild-type PLE, circularization was not detected in the PLE 1 Δ*int* strain. PLE circularization was restored with *in trans* Int expression, but only during ICP1 infection (Fig. 1d, Supplementary Fig. 1a). As Int is necessary for PLE circularization to occur, we next investigated the expression pattern of Int during ICP1 infection. We introduced a FLAG-tag into the endogenous copy of *int* and confirmed that FLAG-tagged Int retained the ability to catalyze circularization within 5 min of ICP1 infection (Supplementary Fig. 1b). We detected FLAG-Int in uninfected cells and observed that the level of Int did not increase during ICP1 infection (Fig. 1e), showing that ICP1 infection does not induce *int* expression. Interestingly, although PLE 1 Δ*int* cannot functionally circularize following phage infection, PLE 1 Δ*int* still inhibits ICP1 plaque formation (Fig. 1f), suggesting that excision is induced separately from other components of PLE that are needed for anti-phage activity.

To determine if PLE 1 Int is a functional LSR, we performed integration assays to probe the ability of Int to recombine the chromosomal *attC* and PLE *attP* sites. Through both in vivo (Supplementary Fig. 1c) and in vitro (Supplementary Fig. 1d) assays, we found that Int was sufficient to recombine *attP* and *attC* sites, as is characteristic of LSRs[22,23]. These assays demonstrate that Int is necessary for PLE circularization in response to ICP1 infection and that Int is a functional LSR that can catalyze recombination between *att* sites.

### PLE requires another factor to direct circularization.
As Int is constitutively expressed and not sufficient to catalyze the circularization of PLE 1 in the absence of phage infection (Fig. 1e, Supplementary Fig. 1a), we hypothesized that an RDF is required to direct Int to recombine the *attL* and *attR* sites as is characteristic of LSRs[25,26]. There are no conserved sequence characteristics of RDFs that enable homology-based identification[27], however, in characterized LSR/RDF systems of temperate phages, both the LSR and RDF are encoded within the same genome[25,28]. To evaluate if the RDF is PLE-encoded, PLE 1 strains harboring gene cluster deletions of predicted open reading frames (ORFs) were screened for circularization defects during ICP1 infection. Unexpectedly, all of the PLE ORF knockouts still circularized, implying that the RDF is not PLE-encoded (Fig. 2a). To establish the minimal PLE-encoded factors required for circularization, we constructed a 'miniPLE', which has Int under control of its endogenous promoter and a kanamycin cassette, flanked by *att* sites, and integrated into the *V. cholerae* chromosome in the same location as PLE 1 (Fig. 2b). In support of the mutational analyses (Fig. 2a), the miniPLE circularized and excised from the chromosome during ICP1 infection (Fig. 2b., Supplementary Fig. 2). Together with the inability of PLE 1 Δ*int* to circularize (Fig. 1d), these results demonstrate that Int is the only PLE gene that is necessary for PLE 1 circularization during ICP1 infection.

### ICP1-encoded PexA is involved in PLE circularization.
Due to the specificity of circularization during ICP1 infection, we hypothesized that ICP1 encodes a gene product that directs Int-

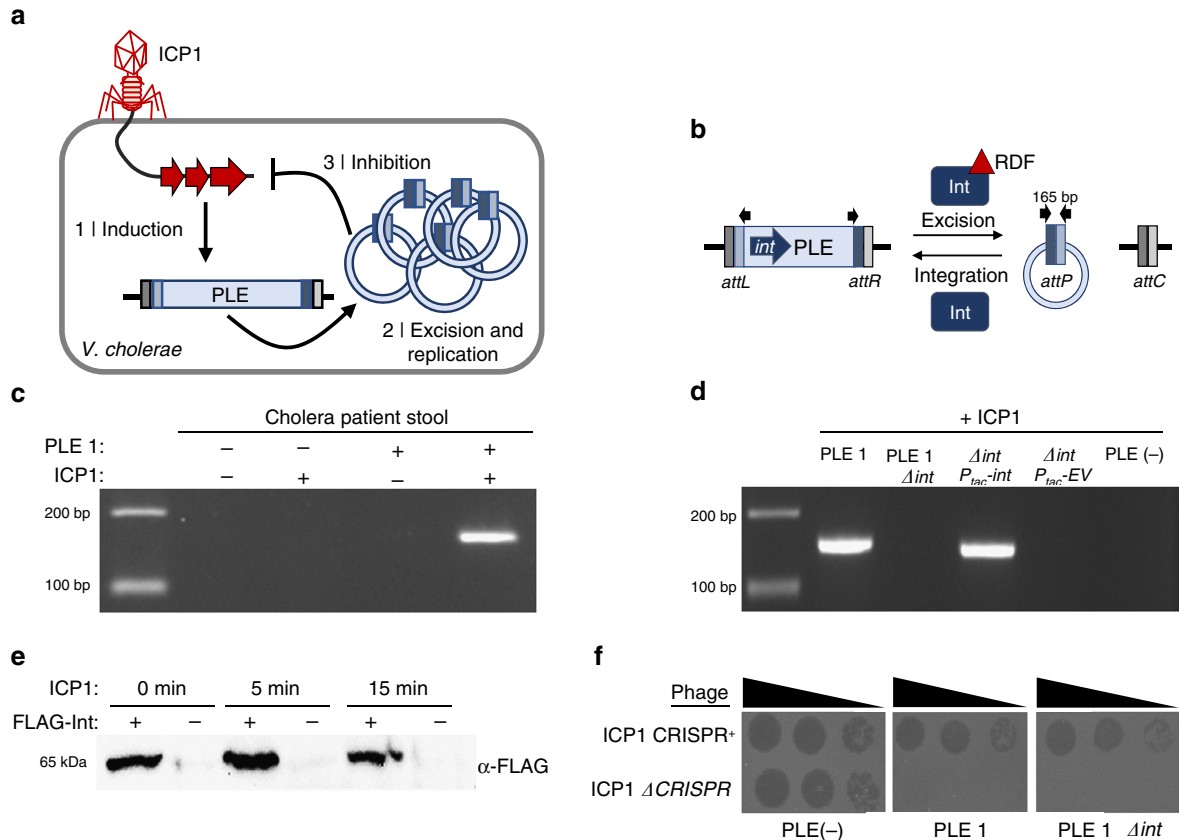

**Fig. 1** PLE circularizes during ICP1 infection and this response is dependent on PLE-encoded Int. **a** ICP1 injects its DNA into a PLE+ *V. cholerae* cell, leading to (1) PLE induction. (2) Induced PLE excises from the chromosome and circularizes, replicating to high copy number[20]. Through an unknown mechanism, (3) PLE inhibits the ICP1 replication program and protects the *V. cholerae* population from ICP1 predation. **b** Model of LSR/RDF mediated integration and excision of PLE and the resulting *att* sites. Black arrows indicate primers used to detect PLE circularization. **c** Circularized PLE, which is detected by PCR using outward-facing primers internal to the PLE, is found in cholera patient stool samples when PLE 1 and ICP1 are both present. **d** Detection of PLE 1 circularization during ICP1 infection. *V. cholerae* with the PLE 1 variant indicated was infected with ICP1 at a multiplicity of infection (MOI) of 5 and harvested 5 min post infection to detect PLE circularization. Strains complemented with a plasmid harboring *int* under control of an IPTG inducible $P_{tac}$ promoter or the empty vector (EV) control were induced 20 min prior to phage infection. **e** Western blot for PLE 1 harboring endogenously FLAG-tagged Int shows Int expression independent of ICP1 infection. Samples of PLE 1 FLAG-Int or untagged PLE 1 Int were collected at the indicated timepoints after infection by ICP1. **f** PLE circularization is not necessary to block ICP1 plaque formation. Tenfold dilutions of ICP1 were spotted onto the listed *V. cholerae* lawns to observe the ability to form plaques (dark spots, zones of killing). ICP1 is unable to form plaques on PLE 1 even in the absence of *int*, while the CRISPR proficient phage is able to overcome PLE 1 and form plaques[18]. Uncropped gels are presented in Supplementary Fig. 5

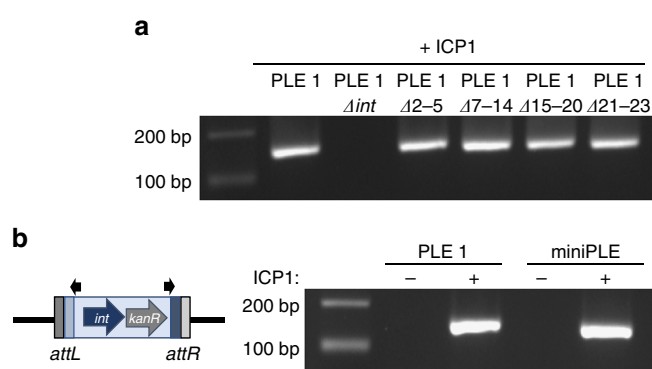

**Fig. 2** PLE 1 does not encode an RDF. **a** Circularization PCR of PLE 1 variants lacking ORFs as indicated, taken 5 min after infection by ICP1 at an MOI of 5. **b** Cartoon of the miniPLE (left) containing PLE 1 *int* and a kanamycin cassette (*kanR*) integrated into the *V. cholerae* chromosome in the same location as PLE 1 with the same *att* sites. Circularization of miniPLE (right) can be detected 5 min post ICP1 infection using the same primers as used for PLE 1 circularization. Uncropped gels are presented in Supplementary Fig. 5

mediated PLE circularization during infection. To identify this phage-encoded gene product, we screened for ICP1 mutations that abolished miniPLE circularization during infection. Through this screen, we identified a mutant phage that failed to circularize the miniPLE (Supplementary Fig. 3a). Sequencing revealed that this mutant phage had a deletion of *orf50* and *orf51* caused by recombination between *orf49* and *orf52* (sites 18761–19800 in ICP1_2004_A, Sequence ID HQ641354 [https://www.ncbi.nlm.nih.gov/nuccore/HQ641354]), leading to a unique fusion gene with a novel stop codon. We evaluated the phage-encoded gene products within this region to determine if one or more was responsible for directing PLE excision during ICP1 infection. Ectopic expression of each gene product revealed that only the hypothetical ICP1 gene product annotated as *orf51* (YP_004250992 [https://www.ncbi.nlm.nih.gov/gene/?term=YP_004250992]) was sufficient to induce PLE circularization in the absence of phage infection (Fig. 3a, Supplementary Fig. 3b). We constructed a clean deletion of *orf51* and found that it was indeed necessary for miniPLE circularization (Fig. 3b, Supplementary Fig 3c), though interestingly, *orf51* is dispensable for ICP1 plaque formation (Supplementary Fig. 3d). As this gene product is both necessary and sufficient for Int-mediated PLE excision, we named it phage-

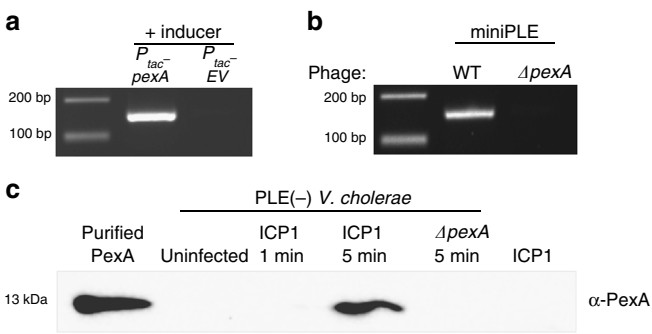

**Fig. 3** ICP1-encoded PexA is necessary and sufficient for PLE cirularization. **a** miniPLE circularizes in the absence of ICP1 infection upon ectopic expression of PexA. **b** miniPLE does not circularize during ICP1 infection when *pexA* is knocked out. **c** Western blot for PexA during ICP1 infection. PLE⁻ *V. cholerae* cultures were probed for PexA at the listed times after infection with ICP1 or ICP1 *ΔpexA*. To determine if PexA is packaged in the phage particle, 5 times the PFU of phage as was used for infection was probed for the presence of PexA. Purified PexA (20 ng) was used as a positive control. Uncropped gels are presented in Supplementary Fig. 5

encoded excisionase (PexA). PexA is a small protein unique to ICP1 isolates that has no sequence similarity to known proteins. Consistent with the rapid kinetics of PLE 1 circularization[20], we found that PexA is expressed de novo within 5 min of ICP1 infection (Fig. 3c), leading us to hypothesize that PexA is hijacked by PLE 1 to function as the RDF for Int-mediated PLE excision.

**PexA is an RDF that directs Int to recombine in vitro.** In order to direct recombination, characterized RDFs physically interact with their cognate LSR. The ability of PexA to physically interact with Int was probed with a bacterial adenylate cyclase two-hybrid (BACTH) assay, in which LacZ expression was detected when both Int and PexA were fused to adenylate cyclase subunits (Fig. 4a). This interaction was further validated using an in vitro pulldown assay, in which PexA coeluted with 6xHis-tagged Int (Fig. 4b), showing that PexA can bind to Int in vivo and in vitro.

Recombination in characterized LSR/RDF systems requires solely the LSR, RDF, and DNA substrates[24,25,28]. To determine if PexA directs Int to excise and circularize PLE, in vitro excision assays were performed using PCR fragments containing the *attR* and *attL* sites and purified PexA and Int (Fig. 4c). Addition of neither PexA nor Int alone led to recombination between *attR* and *attL*; however, when Int and PexA were both added, recombination products were detected (Fig. 4c). RDFs have also been shown to block LSR-mediated integration[25]. Consistent with this model, we observed that addition of PexA to an *attC* and *attP* integration reaction blocks Int-mediated recombination in vitro (Fig. 4d). These data demonstrate that phage-encoded PexA is the RDF for PLE 1 Int and provide the first example of an LSR/RDF pair being encoded in different genomes.

**PexA is conserved and directs excision of historic PLEs.** Analysis of ICP1 genomes from a 12-year period shows that PexA is maintained and that it is 99% identical in all ICP1 isolates (Supplementary Fig. 4a). The conservation of PexA leads us to speculate that although PexA is not essential for plaque formation under lab conditions (Supplementary Fig. 3d), it is likely integral to the ICP1 lifecycle in nature, and hence it may be hijacked as the RDF to signal ICP1 infection in all PLEs. Further, all five PLEs contain a putative LSR and respond to ICP1 infection by circularizing following infection[20], therefore we examined the conservation of the Int/PexA interaction. Upon ectopic expression of PexA, all PLEs, except PLE 2,

demonstrated functional circularization (Fig. 5a). PLE 2 has the most diverse Int, sharing 25.7% amino acid identity with PLE 1 Int across 63% of the protein, while Int from PLE 3, PLE 4, and PLE 5 are more similar to PLE 1 Int (Supplementary Fig. 4b). Consistent with the divergence of PLE 2 Int, PLE 2 also integrates into a unique site in the *V. cholerae* small chromosome[20] (Supplementary Fig. 4c), indicating that PLE 2 Int recognizes different *att* sites from the other PLEs. Altogether, this data indicates that PLE 2 evolved to recognize a unique RDF, possibly altering *att* site specificity in the process.

Although PexA appears to be stably maintained in the natural ICP1 isolates in our collection, the observation that PLE 2 Int has evolved to use a unique RDF indicates that PexA may not have always been a reliable cue enabling PLE to respond to ICP1 infection throughout their co-evolutionary history. We also found that PLE blocks ICP1 infection independent of Int and subsequent circularization (Fig. 1f), suggesting that PLEs may have evolved to use multiple phage products to induce activity. This model of PLE induction is in stark contrast to characterized PICI systems in which a single phage product is sufficient to activate the entire PICI excision–replication–packaging program[8,9]. To test if PexA plays an additional role in PLE activation, we analyzed PLE copy number for PexA-linked effects. During infection with ICP1 *ΔpexA*, PLE still replicated to high copy, although we did observe a mild but not significant defect relative to infection with wild-type phage (Fig. 5b) presumably because PLE is unable to excise from the chromosome and replicate quite as efficiently. Additionally, ectopic expression of *pexA* was not sufficient to drive PLE replication in the absence of ICP1 infection (Fig. 5b), thus PexA functions as the RDF for PLE-encoded Int to stimulate PLE excision but PexA does not appear to play a role in inducing other aspects of PLE activity. Consistent with PexA serving solely as the RDF and our previous observation that PLE *Δint* still blocks ICP1 plaque formation (Fig. 1f), ICP1 *ΔpexA* is still blocked by PLE 1 (Fig. 5c), confirming that PLE circularization is not required for PLE-mediated anti-phage activity.

## Discussion

We report the first LSR/RDF system in which the interacting LSR and RDF are encoded in separate genomes. The exploitation of PexA by PLE demonstrates co-evolution with ICP1, as PLE has evolved to use a conserved phage protein to direct PLE excision and circularization. By constitutively producing Int, *V. cholerae* PLE 1 is perpetually ready to bind PexA, which is expressed early in infection, allowing for rapid response to ICP1 infection. As PexA is unique to ICP1, its function as the RDF for PLE 1 Int defines one component underlying the molecular specificity of the interaction between PLE and ICP1. This interaction has largely persisted, despite the spatiotemporal distribution of samples from which PLEs have been isolated. For example, the oldest *V. cholerae* isolate known to harbor PLE 5 is an Egyptian isolate from 1949[20], yet PLE 5 is specifically mobilized by PexA (Fig. 5a). Conversely, PLE 2 has evolved to recognize a unique RDF while maintaining its response to ICP1. This switch may have been possible because PexA is not essential for ICP1, and although not represented in our collection, it may have been lost in historical epidemics, forcing PLE to co-evolve to continue to recognize persistent ICP1 infection. This specificity and adaptability support the hypothesis that PLE is not a general phage defense mechanism, but an evolved, highly attuned, and specific response evolved to combat continued predation by ICP1.

Since the capacity of PLE to block ICP1 plaque formation is not dependent on PexA-mediated PLE circularization during infection (Fig. 5c), we hypothesize that other, perhaps several other, ICP1-encoded signals induce PLE activity. This model is

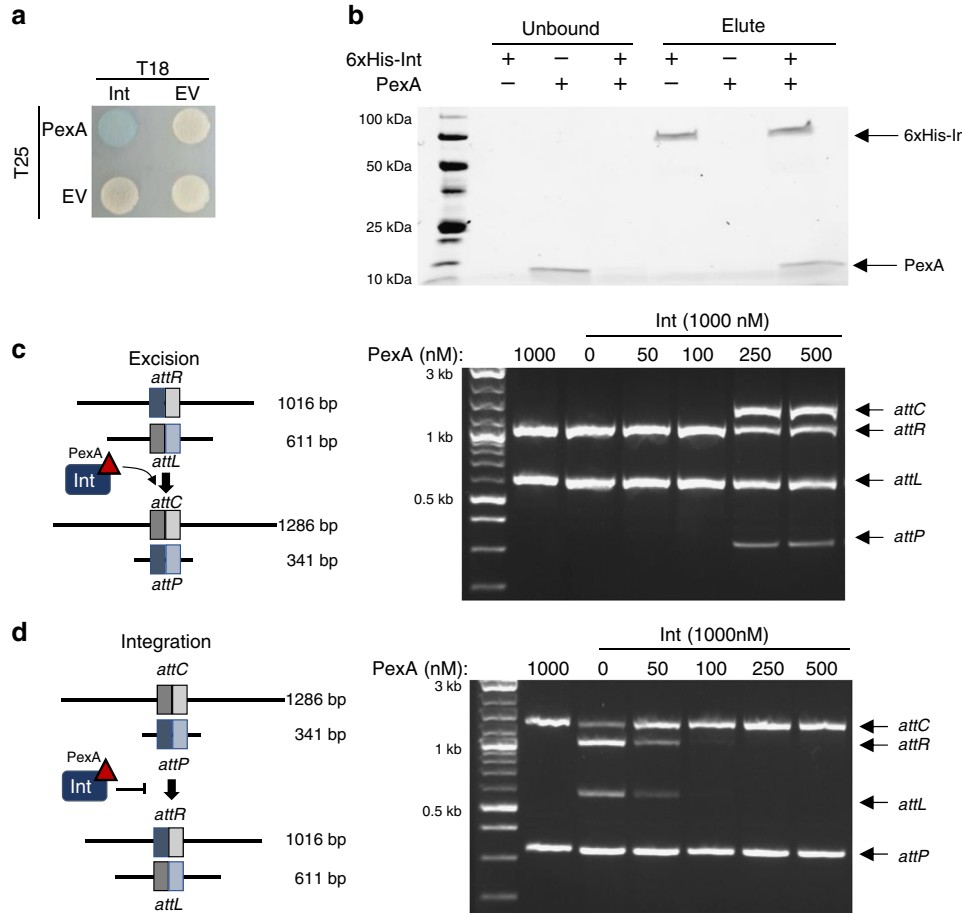

**Fig. 4** PLE exploits phage encoded PexA as the RDF for excision. **a** Bacterial two hybrid analysis to detect protein–protein interactions. Cells containing fusions of PexA to the T25 subunit of adenylate cyclase CyaA (indicated on left) and Int to the T18 subunit (indicated on top) were spotted on X-gal. Blue colonies indicate a physical interaction between the proteins fused to the CyaA subunits. **b** Purified 6xHisSUMO-Int (1 mM) and/or PexA (0.25 mM) were incubated for 30 min and then incubated with nickel resin. Unbound and eluted fractions were collected and were run on an SDS-PAGE gel. **c** Left, cartoon of in vitro excision reaction shows dsDNA fragments containing *attL* and *attR* recombining to *attC* and *attP* in the presence of Int and PexA. Right, in vitro excision assay shows when purified PexA or Int are incubated alone with *attL* and *attR* (lanes 1 and 2, respectively), no recombination is detected. As the concentration of PexA increases, recombination products *attC* and *attP* are detected. **d** Left, cartoon of in vitro integration reaction depicting the ability of PexA to inhibit the activity of Int when recombining *attC* and *attP* sites in an integration reaction. Right, in vitro inhibition of integration (right) by PexA can be seen by the loss of the *attL* and *attR* recombination products as purified PexA increases in concentration. Uncropped gels are presented in Supplementary Fig. 5

divergent from the characterized PICIs in Gram-positive bacteria, in which a single phage protein is necessary to de-represses a PICI master repressor. While PICIs can evolve to recognize a variety of helper phage inducing proteins[29], only a single input is required, leading to induction of the PICI-encoded RDF, and ultimately the entire excision, replication, and packaging cycle[5,8,9]. Accordingly, phages that avoid inducing PICI activity can be selected for in vitro[30]. Not only is the constitutive expression of PLE 1 Int in contrast to the general regulation pattern seen in PICIs and temperate phage alike, our findings also indicate that single evolutionary events that compromise PLE inducing genes in ICP1 cannot prevent PLE induction entirely, thus revealing the evolutionary pressures that lead to the apparent fixation of an active anti-PLE CRISPR–Cas system in ICP1 isolates[18,19].

PLE excision during ICP1 infection enables mobilization of PLE to neighboring cells, and we have previously shown that PLE transduction is Int dependent[20]. It is interesting to note, then, that PLE exploitation of PexA forces ICP1 to directly contribute to horizontal spread of this anti-ICP1 genomic island. Beyond lysogenic phage, serine recombinases have also been shown to have roles in mobilization of pathogenicity islands and antibiotic

resistance cassettes, such as the tandem LSRs that control integration and excision of SCC*mec*, which confers methicillin resistance to *S. aureus*[31]. Since many genes encoded in PLEs have unknown functions and PLE+ *V. cholerae* isolates have been observed with increasing prevalence during recent epidemic sampling[19], it is possible that PLEs may have an underappreciated role in cholera pathogenesis and disease. PLEs confer a known fitness advantage in defending against ICP1 predation[18,20]; however, in light of some of the functional similarities between PLEs and PICIs [20] it is important to consider that PLEs may confer additional traits relevant to the molecular epidemiology of contemporary clones. As such, it is important to understand the mobilization of such islands, which we note, cannot be achieved through genomic analyses alone as there is no sequence-based signature that implicates a role for ICP1 in PLE mobilization. Uncovering the RDF activity of PexA provides insight into the multitude of mechanisms by which newly discovered and even well characterized anti-phage islands can be mobilized by their target phages. Like many lytic phages, ICP1 is being considered for its therapeutic utility[32]; however, without mechanistic insight into phage-bacterial interactions in nature, we may overlook the

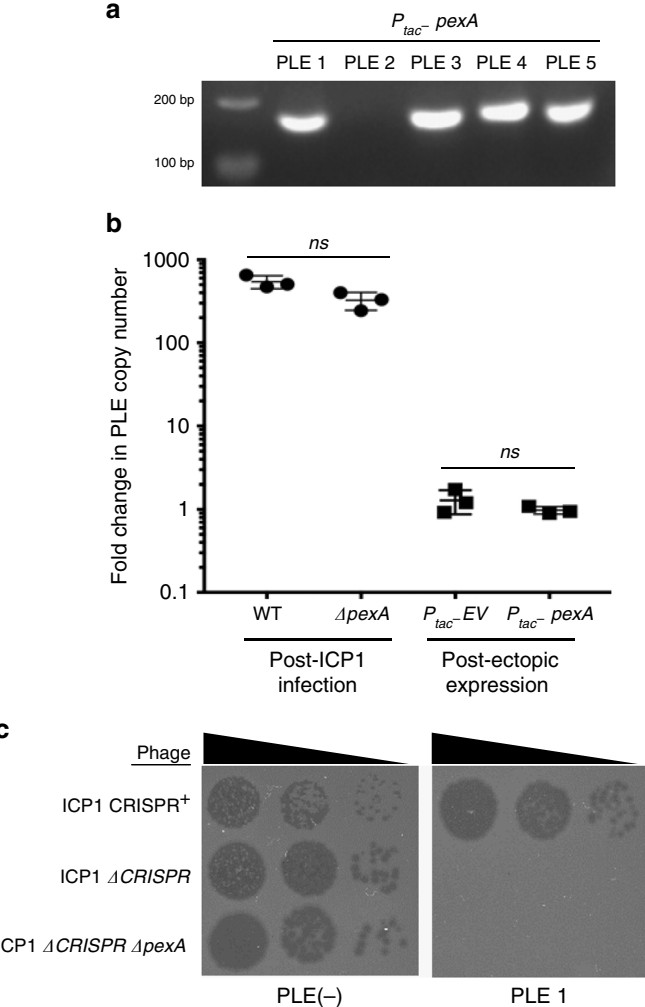

**Fig. 5** PexA directs historic PLEs to circularize and is dispensable for PLE 1 anti-phage activity. **a** Circularization PCR of all PLEs probed during ectopic expression of PexA. **b** Fold change in PLE 1 copy number as measured by qPCR. Left, PLE copy number was compared before and 20 min after ICP1$_{2004}$ or ICP1$_{2004}$ $\Delta pexA$ infection. Right, PLE copy number was examined before and 15 min following ectopic expression of an empty vector (EV) or $pexA$. Bars represent the mean and standard deviation of three independent replicates tested (ns, not significant by T Test). **c** Tenfold dilutions of ICP1 spotted on *V. cholerae* PLE$^-$ or PLE 1 lawns showing the ability of different phage strains to form plaques (dark spots, zones of killing). ICP1 and ICP1 $\Delta pexA$ are unable to form plaques on PLE 1, while the CRISPR proficient phage is able to overcome PLE 1 and form plaques[18]. Uncropped gels are presented in Supplementary Fig. 5

important ways in which ICP1, as well as phage in general, contribute to pathogen evolution.

## Methods

**General growth conditions**. Strains, primers, plasmids, and phage used in this study are listed in Supplementary Tables 1–5. Strains were grown with aeration in LB (lysogeny broth) or on LB agar plates at 37 °C, unless otherwise noted. Where necessary, cultures were supplemented with streptomycin (100 μg/mL), spectinomycin (100 μg/mL), kanamycin (*V. cholerae* 75 μg/mL, *E. coli* 50 μg/mL), ampicillin, (*V. cholerae* 50 μg/mL, *E. coli* 100 μg/mL), chloramphenicol (*V. cholerae* 2.5 μg/mL, *E. coli* 25 μg/mL), or 5-bromo-4-chloro-3-indolyl-beta-D-galacto-pyranoside (X-gal), 40 μg/mL. Ectopic expression constructs in *V. cholerae* were induced with 1 mM IPTG and 1 mM theophylline (P$_{tac}$) or 0.1% arabinose and 1 mM theophylline (P$_{bad}$). Phages were propagated on *V. cholerae* hosts using the soft agar overlay method[20]. High titer phage stocks were collected by polyethylene glycol precipitation[33].

**Strain construction**. PCR products to make chromosomal *V. cholerae* mutants, including chromosomal expression constructs, were created by splicing by overlap extension PCR and introduced by natural transformation[34]. Primer sequences are shown in Supplementary Table 2. Ectopic P$_{tac}$ gene expression vectors were created from a modified pMMB67EH vector[35] or pBAD vector engineered to contain a theophylline inducible riboswitch (Riboswitch E)[36]. Plasmids were constructed using Gibson assembly (NEB), and introduced into *V. cholerae* through conjugation with *E. coli* S17. ICP1 mutants were created as described through CRISPR–Cas gene editing[37]. Briefly, an IPTG-inducible type 1E CRISPR–Cas system was engineered in *V. cholerae* and used to target various regions of the ICP1 genome. Escape phage that are able to form plaques on the targeting host have acquired random mutations in the protospacer region, often as a result of recombination events leading to random deletions. To generate clean deletions, a targeting plasmid was engineered with an additional editing template that served as a recombination template to delete only the gene product targeted by the *V. cholerae* spacer. Successful recombination events allowed ICP1 to form plaques on the targeting host, and all clean deletion constructs were confirmed with DNA sequencing over the region of interest.

**Circularization and excision PCRs**. Stool specimens were collected and stored from previous studies[18]. Total DNA was extracted from 100 μl stool samples using the DNeasy blood and tissue kit (Qiagen), and 2 μL of extracted DNA was used as template to detect PLE, ICP1, and circularized PLE by PCR. For detection of PLE circularization during phage infection, *V. cholerae* strains were grown to OD$_{600}$ = 0.3, infected with phage at a multiplicity of infection (MOI) of 5, and allowed to incubate at 37 °C with aeration. Samples were taken 5 min post infection, boiled for 10 min, and 2 μL was used as template for PCR using primers depicted in Fig. 1b. Resulting reactions were run on a 2% agarose gel and imaged with Gel Green. For Int complementation, ectopic Int was induced for 20 min prior to phage infection. To detect PLE excision, 6 ng genomic DNA from uninfected *V. cholerae* and infected *V. cholerae* harvested 15 min post infection were used as templates for PCR with primers located in the *V. cholerae* chromosome flanking PLE 1. To detect PLE circularization following induction of ectopically expressed PexA, PLE$^+$ derivatives of *V. cholerae* at an OD$_{600}$ = 0.3 were induced for 5 min and samples were boiled and processed for PCR as described above. Images are representative of at least two independent experiments.

**In vivo recombination assay**. Constitutively expressed *lacZ* from *E. coli* was engineered such that *lacZ* was flanked by 300 bp containing *att$_P$* from circularized PLE and 70 bp containing *att$_C$* from a *V. cholerae* repeat (VCR). This construct was integrated into the *V. cholerae* genome at a fixed position (VC2338) by natural transformation. A plasmid with Int or the empty vector control was mated into the reporter strain and individual colonies were picked into 1 mL LB and 2 μL was spotted onto indicator plates with X-gal, IPTG and theophylline.

**PLE circularization screen with mutant phage**. Mutant phages, created by targeting the ICP1 genome with CRISPR and collecting viable escape phage[37], were used to infect the miniPLE *V. cholerae* host using the soft-agar overlay method. Plaques were picked into 50 μL water and boiled for 10 min, and 2 μL of the boiled template was used for circularization PCR as described above. Mutant phage that did not circularize miniPLE were sequenced and, since mutants were not clean deletions of individual gene products, the individual gene products that were missing or mutated were cloned into an inducible vector to test for miniPLE circularization. Strains grown to OD$_{600}$ = 0.3, induced for 20 min, and boiled and processed for PCR as described above. Images are representative of at least two independent experiments.

**Western blot**. *V. cholerae* cultures were grown to OD$_{600}$ = 0.3 and infected with the indicated phage at MOI = 5, and 2 mL samples were taken at the time points indicated. Samples were washed with methanol and phosphate buffered saline (PBS) before being pelleted, re-suspended in 1× Leammli buffer (Bio-rad), and boiled for 10 min. Boiled samples were then loaded on an SDS-PAGE gel for blotting analysis. Rabbit-α-FLAG primary antibodies (Sigma) were used at a dilution of 1:3,000 to detect endogenous FLAG-Int, and custom rabbit-α-PexA primary antibodies (GenScript) were used at a dilution of 1:15,000 to detect PexA. Both were incubated with goat-α-rabbit-HRP conjugated secondary antibodies at a dilution of 1:5000 (Bio-rad). Blots were developed with Clarity Western ECL Substrate (Bio-rad) and imaged on a Chemidoc XRS Imaging System (Bio-rad). Images are representative of at least two independent experiments.

**Quantitative PCR**. To examine PLE replication during phage infection, *V. cholerae* samples were grown to an OD$_{600}$ = 0.3, split into two tubes, and incubated with the listed phage at MOI = 2.5 for 20 min at 37 °C with aeration. Samples were taken just before phage infection and 20 min following infection, boiled for 10 min, and diluted 1:1,000. The qPCR reaction were run using primers Zac14 and Zac15 (Supplementary Table 1) and iQ SYBR Green Supermix (Bio-Rad) and were run on a CFX Connect Real-Time PCR Detection system (Bio-Rad) with the following conditions: denaturation at 95 °C for 5 min, followed by 40 cycles of 95 °C for 10 seconds and 60 °C for 30 seconds. To verify the results, a melt curve analysis was additionally performed. Results were analyzed with 2-tailed paired t test. To

examine the effect of ectopic *pexA* expression on PLE replication, *V. cholerae* strains were grown to an $OD_{600} = 0.3$ and induced for 15 min at 37 °C with aeration. Samples were then taken and boiled for 10 min and compared to samples taken immediately before induction and run as described above. Results were analyzed with a 2-tailed unpaired t test. All experiments were done with three independent biological replicates and each template was quantified in technical duplicate.

**BACTH assay**. Genes of interest were cloned into the multiple cloning sites of pUT18 and pKT25[38] using Gibson Assembly. Three independent colonies were separately picked into 1 mL of LB and 3 μL was spotted onto selective medium containing kanamycin and ampicillin with 0.5 mM IPTG and X-gal. Plates were incubated for 24 h at 30 °C before being imaged.

**Protein preparation**. *E. coli* BL21 cells containing a pE-SUMO fusion to the construct of interest (cloned using Gibson Assembly) were grown in LB with antibiotics at 37 °C to $OD_{600}$~0.9, and induced for 3 h with 0.5 mM IPTG. Cells were centrifuged and resuspended in lysis buffer (50 mM HEPES pH 7.2, 300 mM NaCl, 20 mM imidazole, Pierce™ Protease Inhibitor Mini Tablets (Thermo), 1 mM TCEP, 0.5%Tx-100) and sonicated. Cell debris was removed by centrifugation ($18,000 \times g$ for 40 min at 4 °C), and the lysate was applied to a Nickel resin affinity column (HisPur Ni-NTA Resin). The column was washed with two column-volumes wash buffer (50 mM HEPES pH 7.2, 1 M NaCl, 20 mM Imidizole, 1 mM TCEP) and eluted with elution buffer (50 m M HEPES pH 7.2, 300 mM NaCl, 300 mM Imidizole, 1 mM TCEP). Eluted 6xHisSumo-Int was then run through a HiTrap Heparin HP 5 mL column, and pooled fractions were run on a Superose 6 Increase 10/300 GL column on an AKTA Pure 25 L system (GE Healthcare). Eluted 6xHisSumo-PexA was run on a Superose 6 Increase 10/300 GL column. To cleave the SUMO tag, 1 μL SUMO protease was added per 100 μg of protein and incubated overnight at 4 °C. The mixture was then bound to Novex His-Tag Dynabeads and the unbound fraction was collected and analyzed by SDS-PAGE visualized with *Stain-Free* technology (Bio-rad).

**In vitro pulldown**. Purified 6xHis-SUMO-Int (1000 nM) and/or untagged PexA (250 nM) were added to Novex Dynabeads His-Tag Isolation & Pulldown beads with Binding/Wash buffer (50 mM Sodium-phosphate pH 8, 300 mM NaCl, 0,01% Tween-20) and incubated rocking at room temperature for 10 min. Unbound protein was collected and the resin was washed 4× with Binding/Wash buffer. His Elution buffer (300 mM imidazole, 50 mM sodium-phosphate pH 8, 300 mM NaCl, 0.01% Tween-20) was added, incubated for 5 min rocking at room temperature, and collected. Fractions were analyzed by SDS-PAGE visualized with *Stain-Free* technology (Bio-rad). Images are representative of at least two independent experiments.

**In vitro recombination**. Purified Int and/or PexA were added to 20 μL reactions in the concentrations listed (Fig. [4]c, d, Supplementary Fig. [1]d) with 200 ng purified PCR products containing the indicated *att* sites in buffer (50 mM HEPES pH 7.2, 150 mM NaCL, 10% glycerol, 0.5 mg/mL BSA, 5 mM spermidine). Reactions were incubated for 2 h at 37 °C followed by 10 min at 75 °C to heat inactivate. The entire 20 μL reaction was then run on a 2% agarose gel and visualized with Gel Green. Images are representative of at least two independent experiments.

**Phage plaque spot plates**. *V. cholerae* grown to mid-log was added to 0.7% molten LB agar, poured over a solid agar plate, and allowed to solidify. 3 μL of each ten-fold dilution of phage in LB was spotted onto the solid surface and allowed to dry. Plates were incubated at 37 °C overnight before being imaged. Images are representative of at least two independent experiments.

**Genomic analysis**. Structure prediction of PLE 1 Int was performed using NCBI Conserved Domain Database[39] and the MPI bioinformatics toolkit[40]. PexA sequence was displayed with Weblogo[41]. Alignments for PLE Int were performed with PRALINE[42], and EMBOSS Needle was used to compare PLE 1 and PLE 2 Int[43].

**Data availability**. The data supporting the findings of the study are available in this article and its Supplementary Information Files, or from the corresponding author upon request.

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

## Acknowledgements
The authors would like to thank members of the Seed Lab for thoughtful discussions and feedback. Thanks to Dr. Lindsay Matthews and Dr. Lyle Simmons for the BACTH materials and bacterial strains and plasmids used for protein purification. Thanks to the Glaunsinger Lab for use of their AKTA and specifically Matt Gardner for training on the system. This work was supported by a Kathleen L. Miller Fellowship to ACM from the Henry Wheeler Center for Emerging and Neglected Diseases and grant R01AI127652 to KDS from the National Institute of Allergy and Infectious Diseases (US). KDS is a Chan Zuckerberg Biohub investigator.

## Author contributions
A.C.M. and K.D.S. conceptualized the project, designed experiments and wrote the manuscript. A.C.M. performed the experiments and analyzed the results. K.D.S. supervised the study.

## Additional information

**Competing interests:** The authors declare no competing interests.

