## [Peer Review File · Nature Communications]

Reviewers' comments:

Reviewer #1 (Remarks to the Author):

Understanding how mobile genetic elements (MGEs) interact is absolutely essential in order to decipher the impact that gene transfer has on bacterial virulence and evolution. *Vibrio cholerae* contains a novel family of MGEs, the phage-inducible chromosomal island-like elements (PLEs), the induction and transfer of which depend on a helper phage (phage ICP1). In the current manuscript, the authors demonstrated that the PLE elements encode their own integrase (Int). Expression of this protein is necessary and sufficient for PLE1 integration on the bacterial chromosome. However, although also involved in the process, excision of PLE1 from the bacterial chromosome requires not only the PLE-encoded Int, but also the activity of a recombination directionality factor (RDF), here called PexA (phage-encoded excisionase). While many MGEs (including phages or ICEs) require the activity of both Int and Xis proteins for excision, the novelty presented here is that these proteins are encoded in two different MGEs (the Int on PLE1 and the Xis on the helper phage), which represents a remarkable example of molecular parasitism used by the parasite (the PLE) to sense and respond to the prey (the ICP1).

While the results presented here are potentially interesting for a general audience, this reviewer feels that the story is still at a preliminary stage, and more work should be performed before acceptance of the manuscript. Since many Int/RDF pairs have been extensively characterised previously, the novelty presented here is the different location of the genes encoding the Int and RDF proteins. Additional aspects of the PLE-helper phage interaction should be addressed in terms of proving a much better understanding in how these two elements interact, promoting PLE excision. Specifically:

- Does phage infection induce int expression? If that was the case, which phage-encoded gene controls this? Is the phage-coded pexA responsible for this? In the current model it is clear that the pexA gene is expressed early after phage infection, but nothing is known about the expression of the int gene. The mechanism controlling expression of the PLE int should be included in the manuscript.
- The authors have proposed previously that PLEs replicate to high copy number. It is expected that replication occurs after PLE excision. Has the phage-encoded PexA protein a (direct or indirect) role in inducing PLE replication? Since RDFs can act as transcriptional regulators in other systems, the authors should analyse if the PexA protein could also have additional roles in the PLE cycle. If this was not the case, the authors should show that PLE replication (probably in situ) still occurs after phage infection.

Minor points:

- Page 4, line 78.these results demonstrate that Int is necessary and sufficient for PLE1 circularisation during ICP1 infection. This sentence is not right: Int is necessary but not sufficient; always requires the phage-encoded PexA protein.
- Page 4, line 82. More details about the ICP1 mutants screened and their phenotypes would significantly increase the quality of the manuscript. It is logical that the authors try to keep as much info as possible for future publications, but they should keep in mind that the scientific advance provided in the current manuscript, supporting publication in high impact journal, is questionable.
- Why do the authors call the protein PexA (and not just Pex)? Are other proteins involved in the process? And the name is a bit confusing, because PexA is a PLE excisionase rather than a phage excisionase (although is phage-encoded). Why do they not call the protein as PpxA (phage-encoded

PLE excisionase)?

- Page 5, line 123. Rather than remarkable, the fact that all the PLEs except one (PLE2) respond to ICP1 infection is entirely expected. The authors should clarify that all the PLEs except PLE2 encode the same Int.

Reviewer #2 (Remarks to the Author):

McKitterick and Seed study the specificity of recombination of an recombinase (Int) encoded by a phage-inducible chromosomal island-like elements (PLEs) of *Vibrio cholerae* that displays antagonistic behavior with a lytic virulent phage called ICP1.

Through genetics and biochemistry the authors found that Int targets a ICP1-specific protein, PexA, as a recombination directionality factor (RDF) to sense and excise its genome during ICP1 infection. The authors that this fully function RDF system is encoded by two proteins, on two different genomes. It also provides an explanation for specificity of PLE excision only after phage infection.

This is a cute paper on defining the molecular machinery that leads to PLE excision from the chromosome in response to phage infection. The results suggest also that the 'arms race' between PLEs and phages has been going on for decades and this in turn, argues strongly that lytic phages like ICP1 have been involved in *V. cholerae* ecology for a very long time at least in South Asia.

Comments:

1. Perhaps a bit more can be said about how the phage mutants were made with CRISPR. Were the in-frame deletions, insertions etc.?
2. PexA was identified this way but it is less clear why it came out of a phage-resistance screen. Please explain?
3. The authors say that PexA is non essential but need for phage fitness. Did they have controls of other mutations in the fitness passage experiments? How do they know that the deletion or mutation in PexA is specific? Would any mutation of the same sort show up as less fit even if PLE is still excised?
4. If loss of PexA blocks excision, why does the PLE still have anti-phage effect? I can't remember but I thought blocking PLE excision by deletion of Int will also block its ability to restrict ICP1 replication.

Reviewers' comments:

Reviewer #1 (Remarks to the Author):

Understanding how mobile genetic elements (MGEs) interact is absolutely essential in order to decipher the impact that gene transfer has on bacterial virulence and evolution. *Vibrio cholerae* contains a novel family of MGEs, the phage-inducible chromosomal island-like elements (PLEs), the induction and transfer of which depend on a helper phage (phage ICP1). In the current manuscript, the authors demonstrated that the PLE elements encode their own integrase (Int). Expression of this protein is necessary and sufficient for PLE1 integration on the bacterial chromosome. However, although also involved in the process, excision of PLE1 from the bacterial chromosome requires not only the PLE-encoded Int, but also the activity of a recombination directionality factor (RDF), here called PexA (phage-encoded excisionase). While many MGEs (including phages or ICEs) require the activity of both Int and Xis proteins for excision, the novelty presented here is that these proteins are encoded in two different MGEs (the Int on PLE1 and the Xis on the helper phage), which represents a remarkable example of molecular parasitism used by the parasite (the PLE) to sense and respond to the prey (the ICP1).

We thank the reviewer for taking an interest in our work and for providing valuable feedback to make the manuscript stronger.

While the results presented here are potentially interesting for a general audience, this reviewer feels that the story is still at a preliminary stage, and more work should be performed before acceptance of the manuscript. Since many Int/RDF pairs have been extensively characterised previously, the novelty presented here is the different location of the genes encoding the Int and RDF proteins. Additional aspects of the PLE-helper phage interaction should be addressed in terms of proving a much better understanding in how these two elements interact, promoting PLE excision. Specifically:

- Does phage infection induce int expression? If that was the case, which phage-encoded gene controls this? Is the phage-coded pexA responsible for this? In the current model it is clear that the pexA gene is expressed early after phage infection, but nothing is known about the expression of the int gene. The mechanism controlling expression of the PLE int should be included in the manuscript.

We appreciate the reviewer pointing out an interesting aspect of the PLE-Int story that was not included in the first draft. To address this point, we constructed a tagged PLE-int strain and performed western blot analyses on Int before and during phage infection. We detect constant Int levels in uninfected cells and the levels do not increase upon phage infection (lines 85-89, Fig 1e), indicating that ICP1 infection does not influence *int* expression.

- The authors have proposed previously that PLEs replicate to high copy number. It is expected that replication occurs after PLE excision. Has the phage-encoded PexA protein a (direct or indirect) role in inducing PLE replication? Since RDFs can act as transcriptional regulators in other systems, the authors should analyse if the PexA protein could also have additional roles in

the PLE cycle. If this was not the case, the authors should show that PLE replication (probably in situ) still occurs after phage infection.

We appreciate this suggestion about the potential additional role of PexA in PLE biology and have performed additional experiments that are included in the manuscript: We show that PLE still replicates following infection by ICP1 $\Delta pexA$ (Fig 5b left panel); conversely, we also show that *pexA* expression does not induce PLE replication in the absence of ICP1 infection (Fig 5b right panel). Given that there is no change in inhibition of ICP1 plaque formation on a PLE⁺ host regardless of the presence or absence of *pexA*, we find it unlikely that PexA has any alternative roles in the PLE lifecycle (lines 180-190). We have also highlighted these particular findings in relation to other characterized PICI systems in which a single phage protein is necessary and sufficient to the PICI excision-replication-and packaging cycle in Lines 211-220.

Minor points:

- Page 4, line 78.these results demonstrate that Int is necessary and sufficient for PLE1 circularisation during ICP1 infection. This sentence is not right: Int is necessary but not sufficient; always requires the phage-encoded PexA protein.

We have made this change.

- Page 4, line 82. More details about the ICP1 mutants screened and their phenotypes would significantly increase the quality of the manuscript. It is logical that the authors try to keep as much info as possible for future publications, but they should keep in mind that the scientific advance provided in the current manuscript, supporting publication in high impact journal, is questionable.

We have added more information about the phage screen and the mutant phage that failed to circularize the miniPLE (Supplementary Fig 3a and 3b; lines 123-130). Additionally, we have added more information to the Methods about how the phage mutants were generated (lines 262-271).

- Why do the authors call the protein PexA (and not just Pex)? Are other proteins involved in the process? And the name is a bit confusing, because PexA is a PLE excisionase rather than a phage excisionase (although is phage-encoded). Why do they not call the protein as PpxA (phage-encoded PLE excisionase)?

While the reviewer brings up a good point as to the specificity of the excisionase activity being directed towards PLE, we prefer the name we chose as it is easier to say when verbally describing the work. We also expect to find a second PLE excisionase in the ICP1 genome that is necessary for directing the excision of PLE 2 Int (which we envision naming PexB). Hence, we have elected to keep the naming convention as in the first draft of the manuscript.

- Page 5, line 123. Rather than remarkable, the fact that all the PLEs except one (PLE2) respond to ICP1 infection is entirely expected. The authors should clarify that all the PLEs except PLE2 encode the same Int.

We have made this change.

Reviewer #2 (Remarks to the Author):

McKitterick and Seed study the specificity of recombination of an recombinase (Int) encoded by a phage-inducible chromosomal island-like elements (PLEs) of *Vibrio cholerae* that displays antagonistic behavior with a lytic virulent phage called ICP1.

Through genetics and biochemistry the authors found that Int targets a ICP1-specific protein, PexA, as a recombination directionality factor (RDF) to sense and excise its genome during ICP1 infection. The authors that this fully function RDF system is encoded by two proteins, on two different genomes. It also provides an explanation for specificity of PLE excision only after phage infection.

This is a cute paper on defining the molecular machinery that leads to PLE excision from the chromosome in response to phage infection. The results suggest also that the 'arms race' between PLEs and phages has been going on for decades and this in turn, argues strongly that lytic phages like ICP1 have been involved in *V. cholerae* ecology for a very long time at least in South Asia.

Comments:

1. Perhaps a bit more can be said about how the phage mutants were made with CRISPR. Were the in-frame deletions, insertions etc.?

As a similar request to expand on this section of the manuscript was made by the other reviewer, as such we have added more information about the phage screen and the mutant phage that failed to circularize the miniPLE (Supplementary Fig 3a and 3b; lines 123-130). Additional information has also been added to the Methods section (lines 262-271).

2. PexA was identified this way but it is less clear why it came out of a phage-resistance screen. Please explain?

Our lab has been creating a CRISPR-targeted pool of large, randomized deletions in the ICP1 genome (previously published in Box et al., 2016 *Journal of Bacteriology*. 198:578-590). The mutant phage that failed to induce miniPLE circularization came out of this pool as described in lines 262-271. The present manuscript and the screen described herein did not screen for mutants that resist PLE activity.

3. The authors say that PexA is non essential but need for phage fitness. Did they have controls of other mutations in the fitness passage experiments? How do they know that the deletion or mutation in PexA is specific? Would any mutation of the same sort show up as less fit even if PLE is still excised?

The reviewer raises a valid concern that could be addressed by performing the fitness assay again with a complemented PexA phage. We did attempt to complement the knockout; however, the editing template that contains the PexA reversion template was too toxic to clone

into *V. cholerae* and we were unable to revert the delta PexA phage. As we are unable to construct the appropriate control phage for this experiment, we have withdrawn the fitness competition data from the manuscript. Nonetheless, the comparison of PexA alleles in natural ICP1 isolates from a long time period indicates that PexA is important, so our conclusions have not changed significantly.

4. If loss of PexA blocks excision, why does the PLE still have anti-phage effect? I can't remember but I thought blocking PLE excision by deletion of Int will also block its ability to restrict ICP1 replication.

We thank the reviewer for pointing out this important finding and we have expanded on this point with additional experiments suggested by reviewer 1. We have two figures showing that PLE does not need to excise to have anti-phage activity (Fig 1f for delta Int PLE and Fig 5c for delta PexA ICP1). We also tested for a potential role for PexA in stimulating PLE replication (Fig 5b) and found that PexA is needed only for PLE excision and not for inducing other aspects of PLE activity. We speculate in the discussion that PLE has evolved to use multiple ICP1 inducing cues, which differs significantly from existing paradigms in the PICI literature (lines 209-220). Our data are consistent with the model that PLE maintains a response to ICP1 even with the loss of an individual input, as evidenced by the mutation in *pexA* not being sufficient to escape PLE activity (Fig. 5c, line 189).

REVIEWERS' COMMENTS:

Reviewer #1 (Remarks to the Author):

I have really enjoyed reading this new version of the manuscript. The authors have performed a nice job, and this reviewer feels this is a nice story that deserves to be published in Nature Communications. Just before finishing, just a few minor points:

- in figure 1e, and analysing Int expression, it looks to me that Int expression slightly increases 5 min after phage infection (which matches with PexA expression; Fig. 3a). I do not know if this is reproducible and significant.

- Over the manuscript, it was a very nice comparison among the PICIs and the PLE elements. However, the mechanism by which some SaPIs are induced is a bit more sophisticated than expected. Thus, when the authors say in the discussion (line 210) and in other parts of the manuscript that a single protein serves as the only input to de-repress the SaPIs, a recent paper demonstrated that some SaPIs can be induced by different proteins, all of them performing the same function for the phages (eLife 2017;6:e26487 doi: 10.7554/eLife.26487). These papers (including the current one) highlight the fascinating strategies that some parasites (PICIs and PLEs in this case) use to hijack the phages.

- In some figure legends, western blot is capitalised (the W).

Comments to Reviewer #1

We thank you for the appreciation of the progression of this manuscript. We have addressed the following concerns:

- in figure 1e, and analysing Int expression, it looks to me that Int expression slightly increases 5 min after phage infection (which matches with PexA expression; Fig. 3a). I do not know if this is reproducible and significant.

We appreciate the reviewer's keen eye in analyzing our figures. Any differences between the level of Int at time 0 vs 5 minutes are not able to be detected; however, due to the difficulties in measuring the intensities, we are not making any claims as to significance.

- Over the manuscript, it was a very nice comparison among the PICIs and the PLE elements. However, the mechanism by which some SaPIs are induced is a bit more sophisticated than expected. Thus, when the authors say in the discussion (line 210) and in other parts of the manuscript that a single protein serves as the only input to de-repress the SaPIs, a recent paper demonstrated that some SaPIs can be induced by different proteins, all of them performing the same function for the phages (eLife 2017;6:e26487 doi: 10.7554/eLife.26487). These papers (including the current one) highlight the fascinating strategies that some parasites (PICIs and PLEs in this case) use to hijack the phages.

Thank you for pointing out an additional piece of data to bolster the argument of how elegant these phage defense islands are. The change has been made.

- In some figure legends, western blot is capitalised (the W).

This change has been addressed in the re-working of the manuscript.